# Whac-a-constraint with anomaly-free dark matter models

## Sonia El Hedri[1,★] and Karl Nordström[1,2†]

**1** National Institute for Subatomic Physics (NIKHEF)
Science Park 105, 1098 XG Amsterdam, Netherlands
**2** Laboratoire de Physique Théorique et Hautes Energies (LPTHE),
UMR 7589 CNRS & Sorbonne Université, 4 Place Jussieu, F-75252, Paris, France

★ elhedrisonia@gmail.com, † karl.nordstrom@nikhef.nl

## Abstract

Theories where a fermionic dark matter candidate interacts with the Standard Model through a vector mediator are often studied using minimal models, which are not necessarily anomaly-free. In fact, minimal anomaly-free simplified models are usually strongly constrained by either direct detection experiments or collider searches for dilepton resonances. In this paper, we study the phenomenology of models with a fermionic dark matter candidate that couples axially to a leptophobic vector mediator. Canceling anomalies in these models requires considerably enlarging their field content. For an example minimal scenario we show that the additional fields lead to a potentially much richer phenomenology than the one predicted by the original simplified model. In particular collider searches for pair-produced neutralinos and charginos can be more sensitive than traditional monojet searches in thermally motivated parts of the parameter space where the mediator is outside the reach of current searches.


# 1 Introduction

Dark matter phenomenology has often been studied using so-called Simplified Models [1–8] which aim to consider only the minimal Lagrangian and particle content relevant for relic density calculations, direct and indirect detection, and collider searches. Such models can be understood as a *bottom-up* approach to dark matter model building, to be contrasted with the *top-down* approach focusing on plausible dark matter scenarios in UV-complete theories such as Supersymmetry. However, the requirement for minimality and the focus on phenomenology has also resulted in many of these models failing basic theoretical self-consistency requirements. Models with a new heavy gauge boson, notably, often obfuscate how the masses of the mediator and dark matter are generated in a gauge-invariant manner. Furthermore, models with new fermions that are charged under the Standard Model or under a new gauge group are often plagued by anomalies. Fixing this often requires the introduction of further interactions and fields that will lead to a much richer phenomenology than predicted by the original simplified model [9–19]. It is worth stressing that these problems mean that the models can not be consistently quantised and only provide predictions at tree level, which makes it clear that they are not models of nature in any meaningful sense but rather just benchmarks for specific experimental signatures. Studies of the phenomenology of anomaly-free $U(1)$ extensions in various contexts can be found in [20–42].

Perfect illustrations of this consistency issue are the so-called gauge portal models, involving a fermionic dark matter candidate that is charged under a new dark $U(1)'$ gauge group associated with a massive gauge boson $Z'$. While generating a mass for $Z'$ only requires introducing a new Higgs field, cancelling the anomalies associated with the dark fermion can prove particularly cumbersome. Since these models typically introduce family-independent couplings for the leptons and quarks, they often correspond to $U(1)_B$ and $U(1)_L$ models of gauged baryon and lepton numbers, which were originally discussed in [43–45] with the dark matter phenomenology studied in [46–48]. While simple anomaly-free gauge portal solutions based on this family of models have been discussed in [16], they involve either a large vector-like coupling between the dark matter and the $Z'$, or a sizable coupling between the $Z'$ and the Standard Model (SM) leptons. Both features are somewhat undesirable from a dark matter perspective, as they respectively imply large direct detection cross-sections or a clear dilepton resonance signal at the LHC for large parts of the thermally motivated values of the parameters. Given the current sensitivities for these types of signals [49, 50], it is well-motivated to also investigate anomaly-free models for which the $Z'$ is leptophobic and couples primarily axially to the DM. As outlined in [16, 43] however, anomaly cancellation for these models is non-trivial and requires dramatically increasing the model's particle content. In particular, it is not clear that the mono-X searches which typically are benchmarked with these models at the LHC are the most sensitive once mediator searches are avoided due to the new signatures which are introduced.

In this paper we derive a minimal field content and charge assignment which cancels all gauge anomalies for a model with a fermionic dark matter candidate a vector mediator from a new broken $U(1)_{Y'}$ gauge group. As outlined above, in order to avoid the current direct detection and collider constraints, we require that the mediator is strictly leptophobic, with a purely axial coupling to at least one fermionic interaction eigenstate. We then use this scenario as a benchmark example to illustrate that, in the region of parameter space outside the reach of the dijet searches, anomaly-free gauge portal models will often be more sensitive to searches for new heavy states with electroweak charges at the LHC than to monojet searches. Requiring theoretical consistency for simplified models thus uncovers a particularly intriguing interplay between dark matter searches and the wider BSM programme of the LHC which inspired the

title of this work[1]: constructing a consistent model which avoids the strongest constraints currently set on simplified models predicts new signatures which can allow other searches to exclude the model. This also suggests the usefulness of the Simplified Model as a benchmark for monojet searches is rather questionable. This model was recently independently derived and studied in [19] in the limit where all anomaly-cancelling fermions are decoupled from the dark matter and collider phenomenology, we will here instead focus on the phenomenological consequences of keeping the masses of these new fermions at the same scale as the dark matter fermion.

The structure of the paper is as follows: in Section 2 we derive and discuss the model, with a discussion of the mass mixing scenario we will employ in Section 2.1 and the effect of kinetic mixing of the $U(1)$s in Section 2.2. This is followed by a study of the dark matter phenomenology in Section 3.1 and a study of the LHC phenomenology for a selection of thermal and non-thermal benchmark parameter points in Section 3.2. We conclude in Section 4.

## 2 Model details

We start from a minimal gauge-portal model, with a new Dirac fermion $\chi$ that is charged under a new $U(1)_{Y'}$ gauge group associated with a massive gauge boson $Z'$. The corresponding new physics Lagrangian is

$$\mathcal{L} = -\frac{1}{4}B'^{\mu\nu}B'_{\mu\nu} - \frac{\epsilon}{4}B^{\mu\nu}B'_{\mu\nu} + \bar{\chi}(\displaystyle{\not}\partial - m)\chi - \bar{\chi}(g_V + g_A\gamma_5)\gamma^\mu\chi Z'_\mu, \tag{1}$$

where $B_{\mu\nu}$ and $B'_{\mu\nu}$ are the field strengths for the SM hypercharge and the new $U(1)_{Y'}$ group respectively. In the rest of our study, we set the kinetic mixing $\epsilon$ to zero, briefly discussing this choice in section 2.2. The relative values of the vector and axial couplings $g_{V,A}$ depend on the dark hypercharges of the left and right-handed components of $\chi$, that is, $\chi_L$ and $\chi_R$.

This basic scenario has been widely used to derive constraints from relic density, direct detection, and collider searches for gauge portal models. When the coupling between $\chi$ and $Z'$ is axial, however, which is the configuration with the loosest direct detection constraints, this model has non-zero $U(1)_{Y'}$ anomalies. In what follows, we present an example model with an extended dark sector that allows to cancel these anomalies while keeping the DM-$Z'$ coupling mostly axial and the $Z'$ leptophobic. This corresponds to a specific implementation of the general gauged baryon and lepton-number motivated models derived in [9, 45]. We stress that, although our chosen model has a particularly large number of fields, it is among the most minimal models that can be built with our requirements. A more detailed discussion of the construction of these models is presented by the authors of [16]. In the rest of our study, we will use SARAH [51–56] throughout in order to implement the model and derive relevant quantities.

In order to have an anomaly-free gauge portal model that satisfies the existing experimental constraints, we need to introduce additional fields which transform non-trivially under $U(1)_Y \times SU(2)_L$ [16, 43]. A minimal solution is to add 6 new Weyl fermions in the following $U(1)_Y \times SU(2)_L \times SU(3)_C \times U(1)_{Y'}$ representations[2]:

- $\chi_L \sim (0, \mathbf{1}, \mathbf{1}, Y'_\chi)$,

- $\chi_R \sim (0, \mathbf{1}, \mathbf{1}, -Y'_\chi)$,

---

[1]Whac-a-mole is a classic arcade game in which new moles appear when previous ones have been pushed back into their holes.

[2]An alternative solution with fewer new fields using both $SU(2)_L$ doublets and triplets was derived in [47] and its dark matter phenomenology was studied in [57].

- $\theta_L \sim (Y_\theta, \mathbf{1}, \mathbf{1}, Y'_{\theta_L})$,

- $\theta_R \sim (Y_\theta, \mathbf{1}, \mathbf{1}, Y'_{\theta_R})$,

- $\phi_L \sim (Y_\phi, \mathbf{2}, \mathbf{1}, Y'_{\phi_L})$,

- $\phi_R \sim (Y_\phi, \mathbf{2}, \mathbf{1}, Y'_{\phi_R})$.

Here we have already made $\theta$ and $\phi$ vectorlike under the Standard Model gauge group to avoid having to consider anomaly equations for products of gauge groups not involving $U(1)_{Y'}$. The relevant anomaly equations assuming $Y'_{l,e} = 0$ (which requires $Y'_q = Y'_{u,d}$ to keep the Standard Model Yukawa terms gauge invariant) are then:

$$9Y'_q + Y'_{\phi_L} - Y'_{\phi_R} = 0 \quad SU(2)_L^2 \times U(1)_{Y'}, \tag{2}$$

$$-18Y'_q + 2Y'_{\phi_L}Y_\phi^2 - 2Y'_{\phi_R}Y_\phi^2 + Y'_{\theta_L}Y_\theta^2 - Y'_{\theta_R}Y_\theta^2 = 0 \quad U(1)_Y^2 \times U(1)_{Y'}, \tag{3}$$

$$2Y'^2_{\phi_L}Y_\phi - 2Y'^2_{\phi_R}Y_\phi + Y'^2_{\theta_L}Y_\theta - Y'^2_{\theta_R}Y_\theta = 0 \quad U(1)_Y \times U(1)_{Y'}^2, \tag{4}$$

$$2Y'^3_\chi + Y'^3_{\theta_L} - Y'^3_{\theta_R} + 2Y'^3_{\phi_L} - 2Y'^3_{\phi_R} = 0 \quad U(1)_{Y'}^3, \tag{5}$$

$$2Y'_\chi + Y'_{\theta_L} - Y'_{\theta_R} + 2Y'_{\phi_L} - 2Y'_{\phi_R} = 0 \quad U(1)_{Y'}. \tag{6}$$

In order to avoid having charged dark matter candidates we require $Y_\phi = m + \frac{1}{2}$ and $Y_\theta = n$ where $m, n \in \mathbb{Z}$, so that the lightest charged state can decay into the lightest neutral state by emitting a Standard Model particle. Restricting ourselves to configurations with $|Q| \in \{0, 1\}$ for all new fields we find the solution

$$Y_\theta = \pm 1, \tag{7}$$

$$Y_\phi = \pm\frac{1}{2}, \tag{8}$$

$$Y'_q = -\frac{2}{9}Y'_{\phi_L}, \tag{9}$$

$$Y'_\chi = -Y'_{\phi_L}, \tag{10}$$

$$Y'_{\theta_L} = -Y'_{\phi_L}, \tag{11}$$

$$Y'_{\theta_R} = Y'_{\phi_L}, \tag{12}$$

$$Y'_{\phi_R} = -Y'_{\phi_L}. \tag{13}$$

For simplicity we take $Y'_{\phi_L} = 1$ and $Y_\theta, Y_\phi = 1, \frac{1}{2}$ for the rest of the paper[3].

Due to the gauge invariance of the SM lepton Yukawa interaction term mentioned above, the Standard Model Higgs doublet $H$ must be a singlet under $U(1)_{Y'}$. Therefore in order to break $U(1)_{Y'}$ and give masses to all of the new fields we have to add a new complex scalar which is a Standard Model singlet:

- $\tilde{S} \sim (0, \mathbf{1}, \mathbf{1}, 2)$

This scalar gets a vev $v_S$:

---

[3]While writing up this paper a similar study appeared in [19] which directly connected the model to $U(1)_B$ and hence chose to normalise such that $Y'_q = \frac{1}{3}$, however the models are otherwise identical. We will take a different approach to studying the phenomenology for the remainder of this paper.

$$\tilde{S} = \frac{1}{\sqrt{2}}\left(v_S + S + ia\right),\tag{14}$$

which generates a mass for the $Z'$ gauge boson of $U(1)_{Y'}$

$$m_{Z'}^2 = 4g_{Y'}^2 v_S^2.\tag{15}$$

The allowed mass and interaction terms in the Lagrangian are then as follows:

$$\mathcal{L} \supset -\mu_H^2 H^\dagger H - \mu_S^2 \tilde{S}^2 - \lambda_H |H^\dagger H|^2 - \lambda_{H,S} H^\dagger H \tilde{S}^2 - \lambda_S \tilde{S}^4\tag{16}$$

$$- y_d H \bar{d}q - y_e H \bar{e}l - y_u \widetilde{H}\bar{u}q + h.c.\tag{17}$$

$$- y_{\phi_L,\theta_R}\widetilde{H}\,\overline{\phi_L}\,\theta_R - y_{\phi_L,\chi_R}H\overline{\phi_L}\chi_R - y_{\chi_L,\phi_R}\widetilde{H}\,\overline{\chi_L}\phi_R - y_{\theta_L,\phi_R}H\overline{\theta_L}\phi_R + h.c.\tag{18}$$

$$- y_\chi \tilde{S}\,\overline{\chi_L}\chi_R - y_\theta \tilde{S}\,\overline{\theta_L}\theta_R - y_\phi \tilde{S}^\dagger\overline{\phi_L}\phi_R - y_{\chi_L}\tilde{S}\,\overline{\chi_L}\chi_L^c - y_{\chi_R}\tilde{S}^\dagger\overline{\chi_R}\chi_R^c + h.c.,\tag{19}$$

where $\widetilde{H} = i\sigma^2 H^*$.

## 2.1 From interactions to mass eigenstates

In the broken phase we get the following Majorana mass matrix for the neutral states after expanding the $SU(2)$ doublets as $\phi_{L/R} = \begin{pmatrix}\phi_{L/R,1}\\\phi_{L/R,2}\end{pmatrix}$:

$$\begin{pmatrix}\overline{\chi_L} & \overline{\phi_{R,2}^c} & \overline{\chi_R^c} & \overline{\phi_{L,2}}\end{pmatrix}\begin{pmatrix}\sqrt{2}v_S y_{\chi_L} & 0 & \frac{v_S y_\chi}{\sqrt{2}} & \frac{v_H y_{\chi_L,\phi_R}}{\sqrt{2}}\\ 0 & 0 & \frac{v_H y_{\phi_L,\chi_R}}{\sqrt{2}} & \frac{v_S y_\phi}{\sqrt{2}}\\ \frac{v_S y_\chi}{\sqrt{2}} & \frac{v_H y_{\phi_L,\chi_R}}{\sqrt{2}} & \sqrt{2}v_S y_{\chi_R} & 0\\ \frac{v_H y_{\chi_L,\phi_R}}{\sqrt{2}} & \frac{v_S y_\phi}{\sqrt{2}} & 0 & 0\end{pmatrix}\begin{pmatrix}\chi_L^c\\ \phi_{R,2}\\ \chi_R\\ \phi_{L,2}^c\end{pmatrix}.\tag{20}$$

Turning off the Majorana terms $y_{\chi_L}$ and $y_{\chi_R}$ allows us to describe the propagating degrees of freedom as two Dirac fermions $\chi_{1/2}$ where $\chi_1$ will generically denote the lightest fermion a.k.a. the DM candidate, and that we will always take as mostly $\chi$ to avoid direct detection constraints. Keeping them turned on but small splits the two Dirac pairs into four Majorana fermions [58, 59], which will naturally suppress the diagonal vectorlike couplings of the Majorana dark matter fermion.

The terms mixing the Standard Model singlets with the neutral components of the $SU(2)$ doublets, $y_{\chi_R}$ and $y_{\chi_L}$, also introduce couplings to the $Z$ for $\chi_1$ which again rule the model out through direct detection unless these terms are very small: $y_{\chi_R}, y_{\chi_L} v_H \ll 10^{-2} y_\chi v_S$ for $y_\phi = 1.1 y_\chi$. We keep track of this effect and note that this is a somewhat unfortunate consequence of this particular charge assignment as it requires some fine-tuning to achieve the stated goal of avoiding direct detection constraints. Other anomaly-cancelling solutions such as models involving a $SU(2)_L$ triplet with $Y = 0$ could avoid this hurdle as the neutral fermions would not couple to the $Z$. We present an analysis of the direct detection sensitivity of non-zero $y_{\chi_R}, y_{\chi_L}$ in Section 3.1 to give an estimate of the degree of fine-tuning involved. We note that turning on the Majorana terms in the Lagrangian would also suppress the spin-independent direct detection cross section due to the absence of diagonal vectorlike interactions among the Majorana fermions, and the parameter space and phenomenology would also be more complex due to the presence of four Majorana fermions and a Dirac fermion in the spectrum in the limit where the doublet Yukawa is not much larger than that of the singlet. However this limit

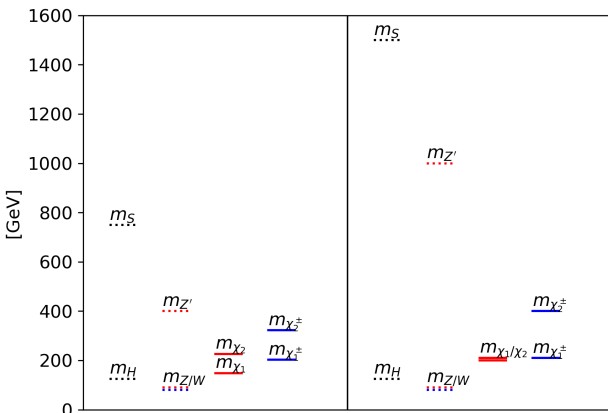

Figure 1: Two representative model spectra which give the correct relic density. The left spectrum shows a scenario with large mixing in both the neutral and charged fermion sectors, and is ruled out by direct detection constraints. The right spectrum is allowed by direct detection and dijet constraints and will be used as benchmark point [3] in Section 3.2.

will share many qualitative features with the same limit in the Dirac case which we explore here[4].

Although the $y_{\chi_R}$ and $y_{\chi_L}$ couplings should be small in order to avoid direct detection constraints, they should also not be identically zero in order to avoid having the neutral component of the doublet $\phi$ —which always couples to the $Z$— being stable, and thus a dark matter candidate. In what follows, we therefore always choose small non-zero values for these couplings in our benchmark models. Note that the existence of these couplings allows $\chi_1$ and $\chi_2$ to coannihilate if they are close enough in mass (so $y_\chi \sim y_\phi$) by ensuring that the two particles remain in equilibrium in the early Universe. We emphasize that studies that involve relic density constraints need to take this possibility into account.

The mass matrix for the charged fermions is:

$$\begin{pmatrix} \overline{\theta_L} & \overline{\phi_{L,1}} \end{pmatrix} \begin{pmatrix} \frac{v_S y_\theta}{\sqrt{2}} & \frac{v_H y_{\theta_L,\phi_R}}{\sqrt{2}} \\ \frac{v_H y_{\phi_L,\theta_R}}{\sqrt{2}} & \frac{v_S y_\phi}{\sqrt{2}} \end{pmatrix} \begin{pmatrix} \theta_R \\ \phi_{R,1} \end{pmatrix}. \tag{21}$$

We denote the two mass eigenstates by $\chi_{1/2}^\pm$. If the off-diagonal elements are small and $y_\theta \gg y_\phi$, $\chi_1^\pm$ is the charged component of the doublet $\phi$, $m_{\chi_1^\pm} \sim m_{\chi_2}$ and $\chi_1^\pm$ can also become a coannihilation partner of the dark matter. We will study this maximal coannihilation scenario in Section 3.1.

For completeness we also include the mixing in the scalar sector in the Appendix A. This mixing has no incidence on the DM phenomenology as long as the contribution of the scalar portal interactions to the DM relic density remain negligible. We will therefore assume $\lambda_{H,S} = 0$ such that the mixing completely vanishes and $H$ is completely Standard Model-like, with the caveat that our results only apply as long as the approximation that any such effects are negligible holds. Including higher order corrections will still induce effective interactions between the dark matter and light quarks through the 125 GeV state, which will in general be constrained

---

[4]In this sense the Dirac scenario provides a good way to study the phenomenology with a smaller parameter space with the understanding that in particular the fine-tuning in the neutral fermion sector mass-mixing would be alleviated by turning on small Majorana terms.

by the spin-independent direct detection cross section, however the $yH\overline{\chi}\chi$ interaction is only constrained to have $y \lesssim 0.1$ by Xenon1T [49] so this mixing is not as fine-tuned as that in the neutral fermion sector.

An interesting question in a model with such a rich particle content is whether there is a decoupling limit for the new anomaly-cancelling fermion states. One possible decoupling direction would be to take the $v_S \to \infty$ limit while requiring $y_\chi$ to be much small than the other Yukawa couplings. We observe, however, that $v_S$ controls not only the fermion masses, but also the mass of the new gauge boson $Z'$ that mediates the dark matter annihilation rate. Notably, in the $m_{\chi_1} > m_{Z'}$ limit where the additional fermions are heavy, most of the dark matter annihilation occurs through the $Z'$, and $y_\chi$ is chosen to maximise this annihilation channel, the associated velocity-averaged cross-section in the $s$-wave verifies

$$\langle \sigma v \rangle \lesssim \frac{g_{Y'}^2}{6 v_S^2}.$$

The observed relic density is achieved for $\langle \sigma v \rangle \approx 10^{-8}$ GeV, which implies that $v_S$ cannot be much larger than 10 TeV for $g_{Y'} \approx 1$. Since, for gauge portal dark matter models, perturbative unitarity sets order one bounds on Yukawa couplings [60], the new fermionic states cannot be made heavier than a few tens of TeV in this limit. In the 'on-shell' region $2m_{\chi_1} < m_{Z'}$ where collider searches are particularly sensitive, it is typically possible to make the anomaly cancelling fermions heavier than the $Z'$ since $m_{Z'} = 2v_S$, $m_{\chi_2}/m_{\chi_1^\pm} \sim y_\phi/\sqrt{2}v_S$ which will heavily suppress their pair production cross section. The simplified model framework for gauge-portal dark matter therefore proves mostly consistent for LHC studies, however as we will show coannihilation effects are well-motivated from a relic density perspective in the on-shell region and does predict clear new LHC signatures.

A potential worry for our model is that since $U(1)_{Y'} \sim U(1)_B$ at low energies, breaking $U(1)_{Y'}$ might induce proton decay or neutron-antineutron oscillations. The general form of the corresponding operator is:

$$\mathcal{O} \propto q^{3p} \ell^q \tilde{S}^r, \tag{22}$$

with $p, q, r \in \mathbb{N}$. Since $Y'_q = 2/9$ and $Y'_{\tilde{S}} = 2$, the minimal gauge invariant realization is given by:

$$\mathcal{O} \propto (q^9 v_S)\ell, \tag{23}$$

with $|\Delta B| = 3$ and mass dimension 16. As such we are not sensitive to proton decay bounds or searches for neutron-antineutron oscillations [61], and any other effect is highly suppressed at $Q \sim \Lambda_{QCD}$ by 12 powers of the scale where $U(1)_{Y'}$ is broken. The general insensitivity of searches for baryon number violation to electroweak scales of $U(1)_B$ breaking when it is gauged has been previously pointed out in e.g. [43, 62, 63].

## 2.2 The effect of kinetic mixing

In general a kinetic mixing term between $U(1)_Y$ and $U(1)_{Y'}$ is allowed by all symmetries:

$$-\frac{\epsilon}{4} B^{\mu\nu} B'_{\mu\nu}. \tag{24}$$

If we remain agnostic about the UV completion of the theory, we can in theory set $\epsilon = 0$ at any scale we wish. However a motivated completion scenario of this model would have the gauge groups unify at some high scale $\Lambda$ which would set $\epsilon = 0$ at that scale, and since the quarks are charged under both $U(1)_Y$ and $U(1)_{Y'}$ and do not have orthogonal charges, this

term would nevertheless run to a non-zero value at the weak scale. The effect on electroweak precision observables for models similar to ours has been studied in [18, 19, 64] which found the resulting constraints to be rather weak. However this mixing will also introduce couplings of the $Z'$ to leptons, as after rotating the kinetic mixing away the general form of the covariant derivative becomes:

$$D_\mu \eta = \left( \partial_\mu - i \sum_{x,y} Q_\eta^x g_{xy} V_\mu^y \right) \eta, \tag{25}$$

where the sum over $x, y = Y, Y'$ denotes the two $U(1)$s. When $\epsilon = 0$, $g_{YY'} = g_{Y'Y} = 0$. A non-zero kinetic mixing will therefore not affect the coupling structure of the dark matter to the $Z'$ but it could make our model dramatically sensitive to LHC searches for dilepton resonances.

In order to show that the effect of the kinetic mixing remains small in most of our parameter space, we have estimated the running of $g_Y$ and $g_{YY'}$ using $g_{Y'}(100 \text{ GeV}) = 0.2, 1$ and $g_Y(m_Z) = 0.358$ when $\epsilon = 0$ at $\Lambda = 10$ TeV. The one-loop renormalisation group equations of these parameters, derived using SARAH [65] (assuming $g_{YY'}, g_{Y'Y} \ll g_Y, g_{Y'}$ and that all new fermions have masses $< m_{Z'}$), are given by:

$$\mu \frac{dg_Y}{d\mu} = \frac{53 g_Y^3}{160\pi^2}, \quad \mu \frac{dg_{Y'}}{d\mu} = \frac{53 g_{Y'}^3}{108\pi^2}, \tag{26}$$

$$\mu \frac{dg_{YY'}}{d\mu} = -\frac{g_Y^2 g_{Y'}}{6\sqrt{15}\pi^2}, \quad \mu \frac{dg_{Y'Y}}{d\mu} = -\frac{g_Y g_{Y'}^2}{6\sqrt{15}\pi^2}. \tag{27}$$

The evolution of the mixing couplings as a function of the energy scale $\mu$ is shown in Figure 2. Calculating di-lepton constraints requires the branching ratio to leptons to be correctly evaluated taking into account $Z' \to \chi\chi$ decays, after which the model-independent cross section limits for narrow resonances in [66] can be recasted after taking into account the production cross section, and the branching fraction into $e^+e^-$ and $\mu^+\mu^-$. The ultimate sensitivity therefore depends on the value of $g_{Y'}$, so ultimately a combination of $g_{Y'}$ and $\Lambda$ will be constrained as demonstrated in Figure 2. The constraint gets weaker for larger values of $m_{Z'}$. According to [19] dilepton constraints become relevant again for thermally motivated values of $g_{Y'}$ for $m_{Z'} \sim \Lambda/100$, but this conclusion only applies when $g_{Y'}$ is fixed by the relic density assuming no coannihilation. In fact, in the coannihilation region, relic density requirements typically point to smaller values of $g_{Y'}$, that lead to a slower running of $g_{YY'}$. In this region, dilepton constraints will therefore be negligible even for smaller values of $m_{Z'}/\Lambda$. Since it is in general always possible to choose $\Lambda$ such that dilepton constraints are negligible we will not consider these when determining the benchmark points which we take as not constrained by dijet constraints in Section 3.2, but a full study of the parameter space should of course specify the assumptions made on $\Lambda$. An interesting study we leave for future work would be to derive a model where $\text{Tr}(YY') = 0$ and the kinetic mixing would not run above the scale where all of the new fermions are dynamical, which would allow the gauge unification scale to be pushed arbitrarily high.

In the following section, we derive the constraints associated with the model described at the beginning of this section. We notably derive the relic density and direct detection constraints for well-motived choices of parameters, and derive and discuss the impact of the current 13 TeV LHC searches for selected benchmark points.

# 3 Phenomenology

This section explores the constraints on our example model from relic density requirements (for benchmark parameter points we require the relic density to have the value measured by

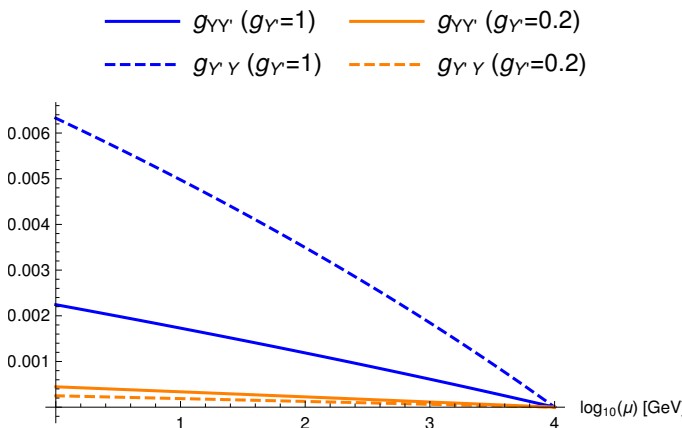

Figure 2: Running of the effective mixing between $U(1)_Y$ and $U(1)_{Y'}$ using the notation introduced in Equation 25 for $\Lambda = 10$ TeV. The $g_{YY'}$ coupling determines the coupling of the mostly $U(1)_{Y'}$ mass eigenstate to leptons.

PLANCK [67]), direct detection, and LHC searches. In order to evaluate these constraints we have implemented our model in SARAH and exported it to the UFO format [68]. The mixing parameters as well as the masses of the different eigenstates are derived from the Yukawa couplings and the vevs using a separate Python code, and directly inputed into the parameter card of the model. We use MG5_AMC@NLO [69] to numerically evaluate the widths of $\chi_2$ and $\chi_{1/2}^{\pm}$ [70] which typically decay into a three-body final state in the coannihilation region. The widths of $Z'$ and $h_2$ are evaluated analytically at tree-level assuming vanishing mixing in the neutral and charged scalar sectors using the general forms given in [6]. Finally, we evaluate the 13 TeV LHC constraints using CHECKMATE 2 [71–83].

### 3.1 Dark Matter Observables

We use MADDM 3.0 [69, 84–86] to calculate the relic density and the direct detection cross-section for our model. We begin by investigating the sensitivity to direct detection experiments induced by the mixing of the DM candidate $\chi_1$ with the neutral components of the new $SU(2)_L$ doublet $\phi$ as discussed above. We find the limits on the spin-independent cross-section with nucleons from Xenon1T [49] to always be the most constraining[5] and plot the resulting constraints as a function of $(y_{\chi_R}, y_{\chi_L})/y_\chi$ for $v_S = v_H$ in Figure 3a. In general these couplings have to verify $y_{\chi_R}, y_{\chi_L} v_H \ll 10^{-2} y_\chi v_S$ for the induced coupling to the $Z$ not to generate a too large cross section when $y_\chi = 1.1 y_\phi$, however this constraint significantly relaxes if there is a hierarchy between $y_\chi$ and $y_\phi$ or $v_H$ and $v_S$. For diagonal Yukawa couplings which are roughly equal, the resulting upper bounds on the mixing between $\chi_1$ and $\chi_2$ still allow for prompt decays of the heavy dark sector particles in the coannihilation region as long as the mass difference is greater than 10 GeV.

Another source of direct detection constraints come from recent constraints on the $\mathcal{O}_{13}$ momentum-dependent operator which appears when integrating out the $Z'$ set by PandaX-II [89]. These constrain a combination of $m_{Z'}$ and $g_{Y'}$ and are stronger than dijet constraints for low $m_{Z'}$. We plot the constraint and the value for some selected parameter points in our model in Figure 3b.

To check that $\chi_2$ and $\chi_1^{\pm}$ can be interconverted with $\chi_1$ efficiently enough for coannihilation to occur we have followed the procedure in [90, 91]. Namely, we compute the rate $\Gamma$ associated

---

[5]The other experimental constraints we make use of are the spin-dependent proton interaction limits set by PICO-60 [87] and the spin-dependent neutron interaction limits set by LUX [88].

with the $t$-channel $\chi_1 q \leftrightarrow (\chi_2, \chi_1^{\pm}) q'$ scattering, that is the dominant $\chi_1$–$\chi_2, \chi_1^{\pm}$ conversion process in our model since the quarks are light. Neglecting the quark masses, the average rate can be expressed as a function of the scattering cross-section $\sigma_{\text{scatt}}$ as

$$\langle \Gamma \rangle = \int_{E_{min}}^{\infty} \sigma_{\text{scatt}} \frac{2 \times 4\pi}{(2\pi)^3} \frac{p_{q,rest}^2 dp_{q,rest}}{e^{p_{q,rest}/T} + 1}, \tag{28}$$

where $p_{q,rest}$ is the momentum of the incoming quark in the rest frame of the heavy initial state fermion. The minimal energy of this incoming quark, $E_{min}$, is defined by

$$E_{min} = \max\left\{ 0, \frac{m_f^2 - m_i^2}{2m_i} \right\} \tag{29}$$

for initial and final states of masses $m_i$ and $m_f$ respectively. We notably find that for $m_{\chi_1} = 150$ GeV, $m_{\chi_2} = 165$ GeV —a relative mass splitting of order 10%— the $q\chi_1 \rightarrow q'\chi_2$ average rate is still many orders of magnitude larger than the Hubble rate at freeze-out for a singlet-doublet mixing approaching the direct detection limit. An example diagram of the process is shown in Figure 4. We therefore consider dark matter masses as low as 150 GeV to be safe when selecting parameter points later in the study, however we display results for $m_{\chi_1}$ below this value in the relic density plots in Figure 5.

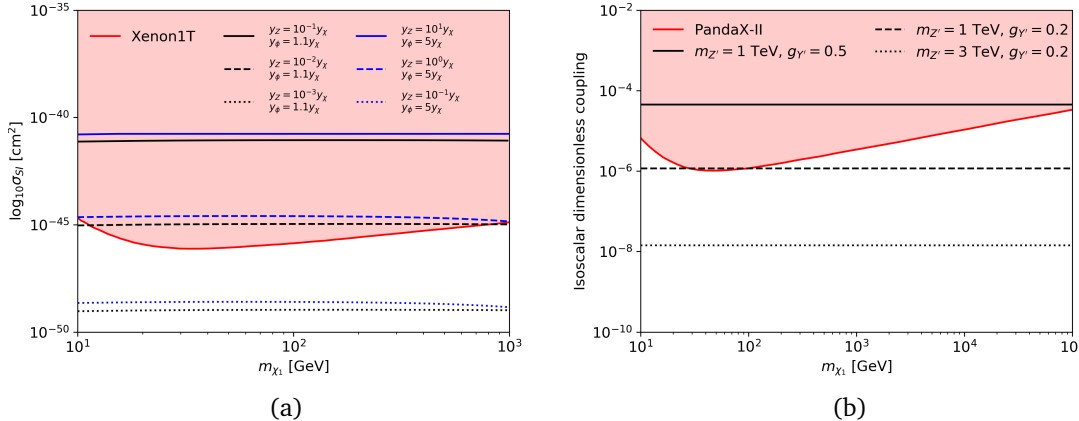

(a)  (b)

Figure 3: The impact of direct detection constraints on (a) the mixing between $\chi_{L,R}$ and the neutral components of the doublets $\phi_{L,R}$ where we have used the shorthand $y_Z = y_{\chi_R}, y_{\chi_L}$ in the legend to indicate that these arise from the coupling to the $Z$. We have set $v_S = v_H$ here, and (b) the mass of the $Z'$ and and $g_{Y'}$ arising from constraints on the $\mathcal{O}_{13}$ operator (here in the form used in [89], where the dimensionless coupling is defined as the squared dimensionless Wilson coefficient when setting the suppression scale to the electroweak vev).

We now compute the DM relic density as a function of $m_{\chi_1}$ and $m_{Z'}$ for a few representative scenarios, showing the results in Figure 5. On one hand we considered a non-coannihilating case, where there is a sizable mass hierarchy between $\chi_1$ and $\chi_2/\chi_1^{\pm}$ and the mixing between the two states does not contribute significantly to the final relic density. This scenario corresponds to that found in typical axial-vector Simplified Models with the addition of diagrams involving $S$, drawn in the top row of Figure 4. On the other hand, we considered two coannihilating scenarios where the $\chi_1 - \chi_2$ and $\chi_1 - \chi_1^{\pm}$ relative mass splittings, defined by

$$\Delta m = \frac{m_{\chi_2/\chi_1^{\pm}} - m_{\chi_1}}{m_{\chi_1}} \tag{30}$$



are fixed to 10% and 2%, and the $\chi_1 - \chi_2$ mixing, albeit small, is sufficient to ensure thermal and chemical equilibrium between the two particles in the early Universe for these values of $\Delta$ (example diagrams of the relevant processes are drawn in the bottom row of Figure 4). We present our results for these three configurations in figure 5. In the non-coannihilating scenario the relic density observed by PLANCK [67] can generically be obtained for perturbative values of all couplings for DM masses around the TeV scale. Setting the DM relic density to its Planck value also allows to uniquely determine the coupling $g_{Y'}$ after $m_{\chi_1}$, $m_{Z'}$, and $m_S$ are fixed. This interesting feature however does not hold when $m_S$ and $\Delta m$ are small enough for coannihilation effects to be significant.

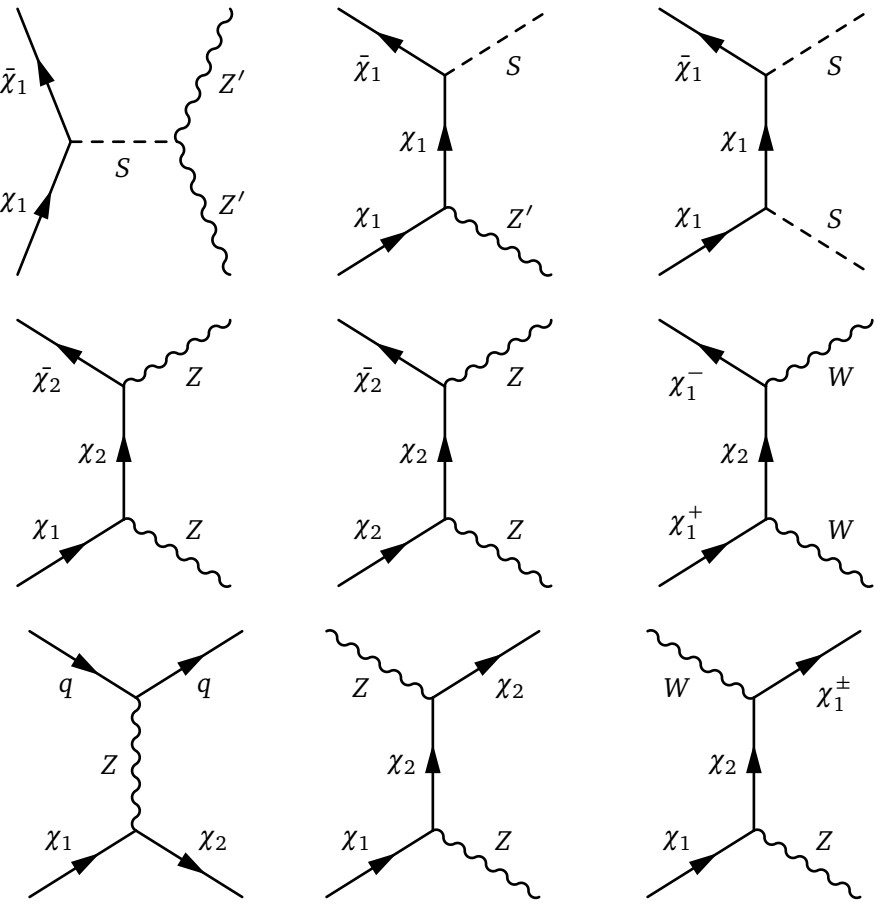

Figure 4: Top row: new diagrams for $\chi_1 \bar{\chi}_1$ annihilation involving $S$ absent in a Simplified Model where the $Z'$ gets its mass using the Stückelberg mechanism. Middle row: example diagrams which contribute to the co-annihilation cross section into weak bosons. Bottom row: example diagrams which allow the conversion processes $\chi_1 \psi \to \chi_2 \psi$ ans $\chi_1 \psi \to \chi_1^{\pm} \psi$ (where $\psi$ is any Standard Model particle) to occur, which are suppressed by one factor of the singlet-doublet mixing. The first process on this row strongly dominates the interconversion rates in the dark sector due to the low quark masses.

Besides the $Z'$ and $S$ funnel regions, where resonance DM annihilation significantly loosens the relic density constraints, we note that the $m_{\chi_1}/2 \ll m_{Z'}$ region always overcloses in the non-coannihilating scenario. Conversely, in the $m_{\chi_1} \gtrsim 2m_{Z'}$ region, the DM relic density decreases when the DM mass increases. This rather counter-intuitive behaviour is due to the fact that this mass is proportional to $v_S$ and $y_\chi$. For a fixed value of $m_{Z'}$ and $g_{Y'}$, with hence

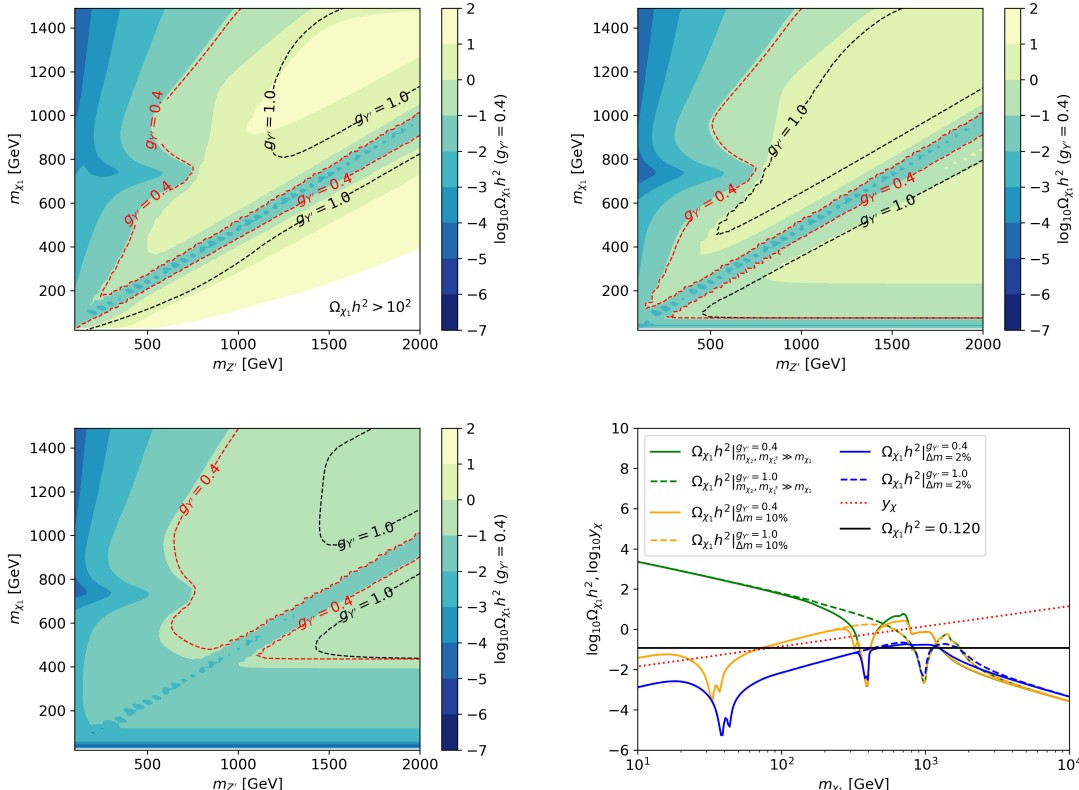

Figure 5: The relic density after freeze-out for $g_{Y'} = 0.4$ as a function of $m_{Z'}$ and $m_{\chi_1}$ for $m_S = 1500$ GeV, $y_\theta = 1.5 y_\chi$. Results are presented as a function of $m_{\chi_1}$ and $m_{Z'}$ for $y_\phi \gg y_\chi$ (no coannihilation, top left), $\Delta m = 10\%$ (top right), $\Delta m = 2\%$ (bottom left), and as a function of $m_{\chi_1}$ for $v_S = 1000$ and $g_{Y'} = 0.4, 1.0$ for these scenarios (bottom right). In the 2D plots the red dashed line shows the contour with the correct relic density for $g_{Y'} = 0.4$, whereas the black dashed line shows the same contour for $g_{Y'} = 1.0$ for comparison.

fixed $v_S$, increasing this mass automatically implies increasing the associated Yukawa coupling, which in turn leads to higher DM annihilation rates through $S$ interactions. We therefore expect perturbativity constraints to set upper limits on the masses of the particles in the dark sector. This is demonstrated in the bottom right plot in Figure 5 where we show the value of $y_\chi$ as a function of $m_{\chi_1}$ for fixed $g_{Y'}, v_S, m_{Z'}$ as a red line. In the coannihilating models with $\Delta m = 2, 10\%$, the shapes of the funnel and $m_{\chi_1} \gtrsim 2 m_{Z'}$ regions are mostly unchanged although the relic density constraints significantly relax at low $\Delta m$. Part of the $m_{\chi_1} \lesssim 2 m_{Z'}$ region, however, is now allowed and the relic density constraints in this area now translate into an upper bound on $m_{\chi_1}$ that does not depend on $m_{Z'}$. This behaviour is due to the fact that the new coannihilation processes that lower the DM relic density in this region are annihilations of dark sector fermions into SM gauge bosons and thus do not involve the $Z'$. These observations are confirmed by the one-dimensional profile of $\Omega h^2$ on the bottom-right plot of figure 5. For the non-coannihilating scenario, the relic density is much larger than its Planck value and overall decreases when the DM mass increases. When coannihilation is significant, on the other hand, the relic density first increases with the DM mass before decreasing again after the $Z'$ funnel. Note also the appearance of the $W$ and $Z$ funnels for coannihilating scenarios where SM electroweak processes dominate at low $m_{\chi_1}$.

Finally, note that indirect detection experiments will also place competitive constraints on

the model, especially in the high-$m_{Z'}$ region (see [11] for a comparative study for a model similar to ours in the limit where all additional fermions are heavy enough to decouple). We will focus on the impact on direct detection, relic density, and LHC constraints from not decoupling the additional fermions in the model here, and leave a detailed study of indirect detection constraints to future work.

## 3.2 LHC Searches

For models that evade the direct detection bounds on the $\chi_1$–$\chi_2$ mixing discussed in section 3.1, the leading experimental constraints are set by colliders, and notably by the 13 TeV LHC searches. Dijet resonance searches are particularly sensitive in the off-shell region where $m_{\chi_1} > m_{Z'}/2$ where the $Z'$ branching ratio to jets is always 100 % [17–19], but only down to $m_{Z'} \sim 450$ GeV, which is the minimal $Z'$ mass probed by $jj$ searches at the LHC, while lower values are probed by $jj +$ ISR searches at the LHC [92–95][6]. Other particularly sensitive probes are direct searches for pair-produced $e^+e^- \to \chi_1^{\pm}\overline{\chi_1^{\pm}}$ charged fermions decaying to a lighter stable neutral fermion $\chi_1$ by emitting a $W$ at LEP. In the MSSM, LEP results rule out charginos lighter than 92.4 GeV in the case where the two charged and neutral fermions are pure doublet Higgsino components [96]. This limit can not be directly translated to our model but suggests that the $Z$ and $W$ funnels that can occur in the co-annihilating case are firmly ruled out.

A third source of collider constraints are monojet searches for $pp \to \chi_1\overline{\chi_1}j$ production where the additional jet comes from ISR. These searches will be competitive at $m_{Z'} < 450$ GeV and $m_{\chi_2/\chi_1^{\pm}} > 100$ GeV where the dijet constraints are weaker and there are no LEP constraints. When the additional fermions in the model are close in mass to $\chi_1$ there are additional $pp \to \chi_2\overline{\chi_2}j\ldots$ processes that also will contribute significantly to the monojet cross section (since the additional decay products of the heavier fermions in the dark sector might be too soft to be picked up by the detector, or invisible in the case of $\chi_2 \to \nu\bar{\nu}\chi_1$) and have to be taken into account.

An additional sensitive channel that has so far been neglected for gauge-portal models is direct searches for pair production of $\chi_2, \chi_1^{\pm}$ at the LHC. In the coannihilation region these particles will often decay to $\chi_1$ promptly by emitting a $Z$ or a $W$ and will therefore have SUSY-like signatures with leptons and jets plus missing transverse energy. The usefulness of searches for these signatures for $U(1)'$ extensions of the MSSM was recently studied in detail in [97]. In general the pair-production cross sections for $\chi_2$ and $\chi_1^{\pm}$ do not depend on $g_{Y'}$ when this coupling is small since SM gauge interactions dominate, which generates a signal which can compete in sensitivity with monojet searches even when the couplings to the $Z'$ are very weak. Note that constraints from these searches can be expected to be most significant when these additional particles are not too heavy. This feature suggests a kind of complementarity between these searches and the relic density constraints since the latter can often be alleviated by bringing the masses of the dark fermions down to the $m_{\chi_2/\chi_1^{\pm}} \sim m_{\chi_1}$ coannihilation region.

To demonstrate the importance of these electroweakino searches and compare them to the monojet searches we use CHECKMATE 2.0.26 to scan all of the implemented 13 TeV searches for the following processes:

- $pp \to \chi_m\overline{\chi_n}j$,

- $pp \to \chi_m\overline{\chi_n}$,

where $\chi_{m/n}$ is summed over all neutral and charged fermions in the dark sector, and the jet is required to have a $p_T > 200$ GeV in order to fill the phase space where monojet searches

---

[6]We use the combination of all of these searches derived in [18] with $gY'_q = \frac{2}{9}g_{Y'}$ in our model to assess these constraints.

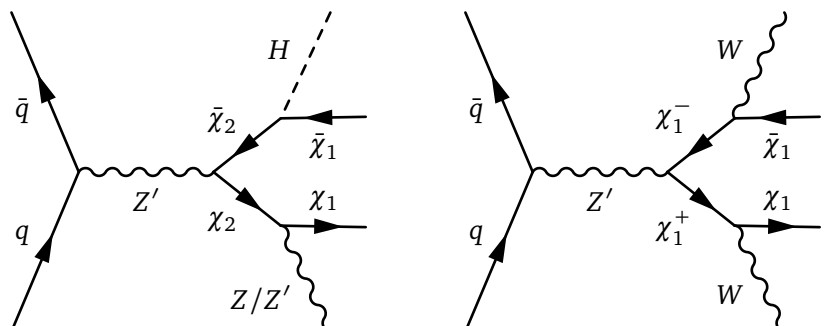

Figure 6: Example pair production processes which contribute to the constraints derived from LHC analyses. Note that the decays of the heavier fermions are three-body if the mass difference to $\chi_1$ is $\ll m_Z/m_{Z'}/m_W$. The branching ratios of the heavier fermions for our benchmark points are given in Tables 1-2.

are sensitive. Some example diagrams contributing to these processes are presented in Figure 6. In general the collider phenomenology is somewhat similar to the Dirac bino – Higgsino limit of the MSSM, with the addition of the $Z'$ which allows for much more efficient s-channel pair production than through the $Z/W$ when $m_{\chi_1} \gg m_Z$. The electroweakino mono-$X$ phenomenology of the MSSM and NMSSM was recently studied in detail in [98] and shares many features with our model.

Due to the computational cost of performing a sensitivity analysis over all the phase space of our model we leave a parameter scan taking all constraints into account to future work and focus on a few benchmark points where we generate $5 \times 10^5$ events per process with minimal phase space cuts. Since many processes often contribute to the same signal region it is necessary to include them all and add up the contributions to derive the strongest constraints possible. Depending on $m_{\chi_1}$ and $\Delta m$ the visible decay products can be either hard —allowing to reconstruct the $W/Z$ masses— or soft, and some three-body decays can be phase-space suppressed and soft compared to others. If the mass splitting is large but pairs of $\chi_2, \chi_1^\pm$ still have non-negligible production cross sections there can be new decay channels opening up, such as $\chi_2 \to h\chi_1$ which can be sensitive to unexpected searches for new signatures like opposite-sign (OS) tau pairs and missing energy. The lifetimes of $\chi_2$ and $\chi_1^\pm$ for the points we consider are shorter than for $b$ mesons in order to be able to consider these prompt. In general smaller mass splitting will further increase the lifetime of the heavier fermions. In this nearly degenerate region where long-lived searches can be expected to be highly sensitive to $\chi_1^\pm$, higher order mass corrections need to be taken into account properly. We leave the analysis of this region to future work, but note that this again suggests the model is sensitive to the wider search program at the LHC for well-motivated parameter choices and that the full mixing scenario needs to be taken into consideration to evaluate these. Here we choose six representative benchmark points that all approximately give the correct relic density within 20% and are summarised in Tables 1,2. These points all escape current collider constraints and show how collider searches provide complementary information in the coannihilation region, however the points in Table 2 are in tension with the latest PandaX-II constraints on the momentum-suppressed spin-independent operator for direct detection. We include them here as examples of possible collider signatures in the model when the direct detection constraints are lifted slightly.

We present the results using the conservative $r$ value for the most sensitive signal regions

Table 1: Summary of benchmark points for scan of LHC searches which can explain the relic density while avoiding all other constraints. In the branching ratios $q$ denotes light quarks and $l$ an electron or muon. The mixing in the neutral fermion sector is set to be small enough to avoid direct detection constraints, which will in general mean it's small enough to be phenomenologically negligible for all other observables except for the lifetimes of the heavier fermions. As an example of these lifetimes, the smallest 10 GeV splitting between $\chi_2$ and $\chi_1$ and $\Delta m = 5\%$ with a mixing within direct detection bounds of points [1], [3] give a $\chi_2$ lifetime of the same order of magnitude as $b$-mesons ($\sim 1.5$ picosecond) and is therefore approaching the limit for where we can ignore searches for long-lived particles, especially for smaller masses where the boosts can be significant.

| Benchmark point | [1] | [2] | [3] |
|---|---|---|---|
| $g_{Y'}$ | 0.6 | 0.3 | 0.2 |
| $m_{Z'}$ | 3000 GeV | 1500 GeV | 1000 GeV |
| $m_S$ | 1500 GeV | 1500 GeV | 1500 GeV |
| $m_{\chi_1}$ | 200 GeV | 300 GeV | 200 GeV |
| $m_{\chi_2}$ | 210 GeV | 312 GeV | 210 GeV |
| $m_{\chi_1^\pm}$ | 210 GeV | 312 GeV | 210 GeV |
| $d_{\chi_2}$ ($\frac{|p|}{m}=1$) | $3.8 \times 10^{-4}$ m | $8.1 \times 10^{-5}$ m | $3.8 \times 10^{-4}$ m |
| $d_{\chi_1^\pm}$ ($\frac{|p|}{m}=1$) | $1.4 \times 10^{-4}$ m | $3.0 \times 10^{-5}$ m | $1.4 \times 10^{-4}$ m |
| $Br(\chi_2 \to \chi_1 q\bar{q})$ | 63% | 61.5% | 63% |
| $Br(\chi_2 \to \chi_1 b\bar{b})$ | 0% | 4.4% | 0% |
| $Br(\chi_2 \to \chi_1 l^+l^-)$ | 8.2% | 7.9% | 8.2% |
| $Br(\chi_2 \to \chi_1 \tau^+\tau^-)$ | 3.3% | 3.4% | 3.3% |
| $Br(\chi_2 \to \chi_1 \nu\bar{\nu})$ | 23.9% | 22.8% | 23.9% |
| $Br(\chi_1^\pm \to \chi_1 q\bar{q})$ | 67.0 % | 66.8% | 67.0% |
| $Br(\chi_1^\pm \to \chi_1 l\bar{\nu}_l)$ | 23.0% | 22.9% | 23.0% |
| $Br(\chi_1^\pm \to \chi_1 \tau\bar{\nu}_\tau)$ | 10.0% | 10.3% | 10.0% |

defined by CHECKMATE,

$$r = \frac{S - 1.64\Delta S}{S95}, \tag{31}$$

where $S$ is the predicted number of signal events, $\Delta S$ is the uncertainty on the signal prediction, and $S95$ is the reported 95% C.L. limit on the BSM cross section in the signal region. Here, we take $\Delta S$ to be solely the statistical uncertainty due to insufficient simulated events and do not consider any theoretical systematic uncertainties or $K$-factors. A point is therefore considered excluded at 95 % C.L. if $r > 1$. Note that since we are only considering models that are allowed by dijet resonance bounds, searches giving $r > 1$ will be the leading collider probe for the corresponding benchmark point.

The results for the four selected parameter points are summarised in Tables 6–8. Points [1] and [3], which achieve the correct relic density through coannihilation processes, are out of the reach of the monojet searches implemented in CHECKMATE but is excluded by searches for opposite-sign (OS) soft lepton pairs and MET [100, 102, 103]. Point [2] which does not have coannihilation is allowed by the LHC searches studied here but is also more sensitive to searches for multiple leptons and MET [108, 109] than to monojet searches. Only point [4] with its large mass splitting between the DM and the other fermions is most sensitive to the monojet search, and even still shows almost comparable sensitivity to a search for opposite-

Table 2: Summary of non-thermal benchmark points for scan of LHC searches. For point [6] we write out the decay modes contributing to the final branching ratio for an experimental signature to demonstrate that larger mass hierarchies in the dark sector means these can no longer be described by (phase-space suppressed) $W/Z$ branching ratios but rather can include cascade decays into $Z'$ and $h$ with modified kinematics, introducing new signatures.

| Benchmark point | [4] | [5] | [6] |
|---|---|---|---|
| $g_{Y'}$ | 0.2 | 0.5 | 0.8 |
| $m_{Z'}$ | 400 GeV | 380 GeV | 240 GeV |
| $m_S$ | 1500 GeV | 1500 GeV | 1000 GeV |
| $m_{\chi_1}$ | 150 GeV | 150 GeV | 60 GeV |
| $m_{\chi_2}$ | 165 GeV | 250 GeV | 600 GeV |
| $m_{\chi_1^\pm}$ | 165 GeV | 250 GeV | 600 GeV |
| $d_{\chi_2}\,(\frac{|p|}{m}=1)$ | $7.7 \times 10^{-6}$ m | $2.1 \times 10^{-9}$ m | $1.9 \times 10^{-11}$ m |
| $d_{\chi_1^\pm}\,(\frac{|p|}{m}=1)$ | $3.0 \times 10^{-6}$ m | $3.6 \times 10^{-10}$ m | $4.3 \times 10^{-11}$ m |
| $\mathrm{Br}(\chi_2 \to \chi_1 q\bar{q})$ | 59.8 % | $\mathrm{Br}(Z \to q\bar{q})$ | $\frac{\mathrm{Br}(Z'\to q\bar{q})}{1.72} + \frac{\mathrm{Br}(Z\to q\bar{q})}{4.76} + \frac{\mathrm{Br}(h\to q\bar{q})}{5.0}$ |
| $\mathrm{Br}(\chi_2 \to \chi_1 b\bar{b})$ | 7.8 % | $\mathrm{Br}(Z \to b\bar{b})$ | $\frac{\mathrm{Br}(Z'\to b\bar{b})}{1.72} + \frac{\mathrm{Br}(Z\to b\bar{b})}{4.76} + \frac{\mathrm{Br}(h\to b\bar{b})}{5.0}$ |
| $\mathrm{Br}(\chi_2 \to \chi_1 l^+l^-)$ | 7.4 % | $\mathrm{Br}(Z \to l^+l^-)$ | $\frac{\mathrm{Br}(Z\to l^+l^-)}{4.76} + \frac{\mathrm{Br}(h\to l^+l^-)}{5.0}$ |
| $\mathrm{Br}(\chi_2 \to \chi_1 \tau^+\tau^-)$ | 3.4 % | $\mathrm{Br}(Z \to \tau^+\tau^-)$ | $\frac{\mathrm{Br}(Z\to \tau^+\tau^-)}{4.76} + \frac{\mathrm{Br}(h\to \tau^+\tau^-)}{5.0}$ |
| $\mathrm{Br}(\chi_2 \to \chi_1 \nu\bar{\nu})$ | 21.6 % | $\mathrm{Br}(Z \to \nu\bar{\nu})$ | $\frac{\mathrm{Br}(Z\to \nu\bar{\nu})}{4.76}^{\dagger}$ |
| $\mathrm{Br}(\chi_1^\pm \to \chi_1 q\bar{q})$ | 67.0 % | $\mathrm{Br}(W \to q\bar{q})$ | $0.98 \times \mathrm{Br}(W \to q\bar{q})^{\dagger\dagger}$ |
| $\mathrm{Br}(\chi_1^\pm \to \chi_1 l\bar{\nu}_l)$ | 22.5 % | $\mathrm{Br}(W \to l\bar{\nu}_l)$ | $0.98 \times \mathrm{Br}(W \to l\bar{\nu}_l)^{\dagger\dagger}$ |
| $\mathrm{Br}(\chi_1^\pm \to \chi_1 \tau\bar{\nu}_\tau)$ | 10.5 % | $\mathrm{Br}(W \to \tau\bar{\nu}_\tau)$ | $0.98 \times \mathrm{Br}(W \to \tau\bar{\nu}_\tau)^{\dagger\dagger}$ |

$\dagger$: there is also an invisible $\mathrm{Br}(Z' \to \chi_1\overline{\chi_1})$ contribution which is experimentally indistinguishable.

$\dagger\dagger$: the total branching ratios are slightly rescaled due to rare $\chi_1^\pm \to \chi_1 t\bar{b},\ \chi_1 hW,\ \chi_1 ZW,\ \chi_1 Z'W$ three-body decays.

Table 3: Summary of constraints on benchmark point [1] from the most sensitive 13 TeV searches implemented in CHECKMATE at the LHC. The two last results where a similar analysis with less data is more sensitive can be explained by differences in the base selection criteria. All the signal regions shown here require at least $\not{E}_T \gtrsim 100$ GeV. SRI-MLL-10 is inclusive in lepton flavor and select lepton pairs with invariant mass $m_{\ell\ell} \in [1,20]$ GeV. EM2 simply requires $\not{E}_T \in [300,350]$ GeV. Finally, the last region targets relatively compressed spectra and cuts on the leading lepton $p_T$.

| Analysis | Constraint on [1] |
|---|---|
| ATLAS 36.1 fb$^{-1}$ jets + MET [99] | $r = 0.08$, 2j-2100 |
| ATLAS 36.1 fb$^{-1}$ soft OS lepton pair + MET [100] | $r = 0.33$, SRI-MLL-10 |
| ATLAS 36.1 fb$^{-1}$ $j$ + MET [101] | $r = 0.12$, EM10 |
| CMS 12.9 fb$^{-1}$ soft OS lepton pair + MET [102] | $r = 0.061$, stop low MET low $p_{T,l_1}$ |
| CMS 35.9 fb$^{-1}$ soft OS lepton pair + MET [103] | $r = 0.049$, weakino low MET high $m_{ll}$ |

Table 4: Summary of constraints on benchmark point **[2]** from the most sensitive 13 TeV searches implemented in CHECKMATE at the LHC. The EM2 region is the same as the one shown in Table 6. $SR_W^{3-body}$-SF is originally designed for stops with $\Delta m(\tilde{t}, \chi) \sim m_W$ and hence imposes a $b$-quark veto and cuts on a "super-Razor" variable $M_\Delta^R$, that reaches an endpoint near the stop-neutralino mass splitting. 3LI simply imposes a moderate missing $E_T$ cut and a tight requirement on the $p_T$ of the third lepton. Finally, SR A25 imposes a harder $\not{E}_T$ cut as well as a transverse mass cut, and tags $Z$ bosons.

| Analysis | Constraint on [2] |
|---|---|
| ATLAS 36.1 fb$^{-1}$ soft OS lepton pair + MET [100] | $r = 0.66$, SRI-MLL-20 |
| ATLAS 14.8 fb$^{-1}$ isolated lepton + jets + MET [104] | $r = 0.12$, GG2J |
| ATLAS 36.1 fb$^{-1}$ $j$ + MET [101] | $r = 0.45$, EM10 |
| CMS 12.9 fb$^{-1}$ soft OS lepton pair + MET [102] | $r = 0.21$, stop low MET low $p_{T,l_1}$ |
| CMS 35.9 fb$^{-1}$ soft OS lepton pair + MET [103] | $r = 0.18$, stop medium MET high $p_{T,l_1}$ |

Table 5: Summary of constraints on benchmark point **[3]** from the most sensitive 13 TeV searches implemented in CHECKMATE at the LHC. SRI1, EM2, and the stop low MET low $p_{T,\ell_1}$ regions are already described in Table 6. 2j-1200 imposes a high MET cut , requires at least two hard jets with invariant mass larger than 1.2 TeV. SRI-MLL-10 is inclusive in lepton flavour and select lepton pairs with invariant mass $m_{\ell\ell} \in [1, 10]$ GeV. GG2j is similar to the regions used in monojet searches but also requires leptons. Finally, the weakino high MET low $p_{T,\ell_1}$ region requires a same-flavour opposite sign lepton pair and cuts on its invariant mass.

| Analysis | Constraint on [3] |
|---|---|
| ATLAS 36.1 fb$^{-1}$ $\gamma$ + MET [105] | $r = 0.010$, SRI1 |
| ATLAS 36.1 fb$^{-1}$ jets + MET [99] | $r = 0.024$, 2j-1200 |
| ATLAS 36.1 fb$^{-1}$ soft OS lepton pair + MET [100] | $r = 1.3$, SRI-MLL-10 |
| ATLAS 14.8 fb$^{-1}$ iso lepton + jets + MET [104] | $r = 0.033$, GG2J |
| ATLAS 36.1 fb$^{-1}$ $j$ + MET [101] | $r = 0.24$, EM2 |
| CMS 12.9 fb$^{-1}$ soft OS lepton pair + MET [102] | $r = 0.17$, stop low MET low $p_{T,l_1}$ |
| CMS 35.9 fb$^{-1}$ soft OS lepton pair + MET [103] | $r = 0.15$, weakino high MET low $p_{T,l_1}$ |

sign $\tau$s + MET in the high $m_{\tau\tau}$ signal region ($m_{\tau\tau} > 110$ GeV) [106], which can be traced back to the 20% branching fraction of $\chi_2 \to h\chi_1$.

These results demonstrate that monojet searches often do not provide the only or most sensitive collider constraints on anomaly-free gauge portal models which do not couple to leptons in the part of parameter space where dijet constraints are not applicable. Viable coannihilation scenarios, for example, are often particularly sensitive to searches for compressed SUSY spectra. These scenarios, with their compressed topologies and relaxed relic density constraints are expected to gain importance as collider and direct detection constraints become even more stringent in the rest of the parameter space[7]. Even in scenarios without coannihilation it is possible to find interesting new signatures due to the heavier fermions if they are not completely decoupled, which often is necessary to avoid nonperturbative Yukawas after fixing the other masses in the model. Thus, even seemingly simple scenarios such as portal models are

---

[7]Of course direct detection constraints can also eventually push the allowed mixing in the fermion sector to be too small for coannihilation to occur.

Table 6: Summary of constraints on benchmark point [4] from the most sensitive 13 TeV searches implemented in CHECKMATE at the LHC. The two last results where a similar analysis with less data is more sensitive can be explained by differences in the base selection criteria. All the signal regions shown here require at least $\not{E}_T \gtrsim 100$ GeV. The SRI1 signal region selects events with no more than two jets and a hard photon. The SR2 highmass region is designed for events with two hard taus, with large invariant mass and $m_{T2}$. SRI-MLL-30 is inclusive in lepton flavor and select lepton pairs with invariant mass $m_{\ell\ell} \in [1,30]$ GeV. EM2 simply requires $\not{E}_T \in [300,350]$ GeV. Finally, the last region targets relatively compressed spectra and cuts on the leading lepton $p_T$.

| Analysis | Constraint on [4] |
| --- | --- |
| ATLAS 36.1 fb$^{-1}$ $\gamma$ + MET [105] | $r = 0.021$, SRI1 |
| ATLAS 36.1 fb$^{-1}$ OS $\tau$ pair + MET [106] | $r = 0.058$, SR2 highmass |
| ATLAS 36.1 fb$^{-1}$ soft OS lepton pair + MET [100] | $r = 1.2$, SRI-MLL-30 |
| ATLAS 36.1 fb$^{-1}$ $j$ + MET [101] | $r = 0.12$, EM2 |
| CMS 12.9 fb$^{-1}$ soft OS lepton pair + MET [102] | $r = 3.3$, stop low MET low $p_{T,l_1}$ |
| CMS 35.9 fb$^{-1}$ soft OS lepton pair + MET [103] | $r = 1.1$, stop low MET low $p_{T,l_1}$ |

Table 7: Summary of constraints on benchmark point [5] from the most sensitive 13 TeV searches implemented in CHECKMATE at the LHC. The EM2 region is the same as the one shown in Table 6. $SR_W^{3-body}$-SF is originally designed for stops with $\Delta m(\tilde{t}, \chi) \sim m_W$ and hence imposes a $b$-quark veto and cuts on a "super-Razor" variable $M_\Delta^R$, that reaches an endpoint near the stop-neutralino mass splitting. 3LI simply imposes a moderate missing $E_T$ cut and a tight requirement on the $p_T$ of the third lepton. Finally, SR A25 imposes a harder $\not{E}_T$ cut as well as a transverse mass cut, and tags $Z$ bosons.

| Analysis | Constraint on [5] |
| --- | --- |
| ATLAS 13.3 fb$^{-1}$ 2 OS leptons + 2 $bs$ + MET [107] | $r = 0.10$, $SR_W^{3\text{-body}}$-SF |
| ATLAS 13.3 fb$^{-1}$ 2,3 leptons + MET [108] | $r = 0.15$, 3LI |
| ATLAS 36.1 fb$^{-1}$ $j$ + MET [101] | $r = 0.046$, EM2 |
| CMS 35.9 fb$^{-1}$ 2,3 leptons + MET [109] | $r = 0.14$, SR A25 |

Table 8: Summary of constraints on benchmark point [6] from the most sensitive 13 TeV searches implemented in CHECKMATE at the LHC. The SR2 highmass and EM2 regions have been described in Table 6. The 4j-1400 region imposes a hard cut on missing $E_T$, requires at least four hard jets with invariant mass larger than 1.4 TeV.

| Analysis | Constraint on [6] |
| --- | --- |
| ATLAS 36.1 fb$^{-1}$ OS $\tau$ pair + MET [106] | $r = 0.31$, SR2 highmass |
| ATLAS 36.1 fb$^{-1}$ jets + MET [99] | $r = 0.18$, 4j-1400 |
| ATLAS 36.1 fb$^{-1}$ $j$ + MET [101] | $r = 0.49$, EM2 |

in fact associated with a wide variety of collider searches and a full study of these models requires the scan of as many published analyses and signal regions as possible, which mirrors the situation in for example the pMSSM. Finally, we note that aside from SUSY searches, studying

constraints from Standard Model measurements using CONTUR could also lead to meaningful constraints on this class of models [110].

Finally, let us point out that if we loosen the relic density constraint to also allow relic densities which are smaller than the measured one, we will in general require larger $g_{Y'}$, which suggests even more sensitivity to LHC searches (assuming the direct detection constraint on the momentum-suppressed spin-independent operator still is avoided).

# 4 Conclusions

Simplified dark matter models where a fermion annihilates into SM particles via a vector mediator are popular benchmark scenarios. In order to avoid very strong direct detection and dilepton resonance constraints in these models it is convenient to keep the coupling of the vector boson to the DM candidate axial and make it leptophobic. This choice of parameters, however, introduces gauge anomalies when the vector is the gauge boson of a new broken $U(1)$ gauge group. We have demonstrated that the phenomenological consequences of avoiding such anomalies by enlarging the field content of the model can be wide-reaching even for the simplest possible solution. The additional particles allow for much richer scenarios than in simplified models, due to the presence of new weakly charged fermions and the possibility of coannihilation. It follows that constraints on simplified dark matter models which assume leptophobia and an axial DM–$Z'$ coupling should be understood as applying only to a specific corner of the full parameter space of the minimal realistic model. We have in effect played a game of Whac-A-Mole with experimental constraints, avoiding some but in the process introduced several new ones. In particular it is humorous that constructing a leptophobic model to avoid dilepton resonance constraints predicts new signatures for searches for opposite-sign lepton pairs. Notably, regions of the parameter space of the anomaly free theory can be better constrained using LHC searches for electroweakinos rather than monojet searches. These results further highlight the value of the full width of the BSM program at the LHC from a dark matter perspective.

Finally, we note that the anomaly-free model studied here is only one example of how to extend gauge portal simplified models to make them theoretically consistent. A wider study of how to cancel anomalies in $Z'$ models without running into direct detection or LHC dilepton resonance constraints and the typical additional constraints that one should expect would be particularly useful. Additionally, an in-depth study of some of the more exotic signatures associated with our scenario such as long-lived particles would be of prime importance to fully understand this class of models.

Although simplified models proved incredibly useful tools for exploring new physics model at colliders and dark matter experiments, it is crucial to keep their limitations in mind and question their minimality. The example of anomaly cancellation illustrates that even enforcing basic consistency requirements on these models can considerably enlarge the associated phenomenology, with a wide palette of new signatures to explore.

# A Appendix

The scalar mass matrix is:

$$
\begin{pmatrix} h & s \end{pmatrix}
\begin{pmatrix}
3\lambda_H v_H^2 + \frac{\lambda_{H,S} v_S^2}{2} + \mu_H^2 & \lambda_{H,S} v_H v_S \\
\lambda_{H,S} v_H v_S & \frac{\lambda_{H,S} v_H^2}{2} + 3\lambda_S v_S^2 + \mu_S^2
\end{pmatrix}
\begin{pmatrix} h \\ s \end{pmatrix}.
\tag{32}
$$

The tadpole equations are:

$$\mu_H^2 v_H + \lambda_H v_H^3 + \frac{1}{2}\lambda_{H,S} v_H v_S^2 = 0\,,$$

(33)

$$\mu_S^2 v_S + \frac{1}{2}\lambda_{H,S} v_H^2 v_S + \lambda_S v_S^3 = 0\,.$$

(34)

Using these we get the following physical masses for the two scalar mass eigenstates $h_{1/2}$ (where we will use $h_1$ as the 125 GeV state):

$$m_{h_{1/2}}^2 = \left\{ \lambda_H v_H^2 + \lambda_S v_S^2 \pm \sqrt{\lambda_H^2 v_H^4 + \lambda_{H,S}^2 v_H^2 v_S^2 - 2\lambda_H \lambda_S v_H^2 v_S^2 + \lambda_S^2 v_S^4} \right\}\,.$$

(35)

The pseudoscalar mass mixing matrix is:

$$\begin{pmatrix} G^0 & a \end{pmatrix} \begin{pmatrix} \begin{array}{c} \mu_H^2 + \lambda_H v_H^2 + \frac{\lambda_{H,S} v_S^2}{2} \\ + \frac{1}{4} g_2^2 v_H^2 \cos^2\theta_W R_\xi^Z + \frac{1}{2} g_1 g_2 v_H^2 \cos\theta_W R_\xi^Z \sin\theta_W \\ + \frac{1}{4} g_1^2 v_H^2 R_\xi^Z \sin^2\theta_W \end{array} & 0 \\ 0 & \begin{array}{c} 4 g_{Y'}^2 v_S^2 R_\xi^{Z'} + \frac{\lambda_{H,S} v_H^2}{2} \\ + \lambda_S v_S^2 + \mu_S^2 \end{array} \end{pmatrix} \begin{pmatrix} G^0 \\ a \end{pmatrix}\,.$$

(36)

The absence of mixing is due to our scalars $H$ and $S$ breaking $U(1)_Y \times SU(2)_L$ and $U(1)_{Y'}$ separately. We can use the tadpole equations to determine the masses of the pseudoscalars $A_{1/2}$ (which of course are eaten by $Z$ and $Z'$ in unitary gauge) as above:

$$m_{A_{1/2}}^2 = \left\{ \frac{1}{4} v_H^2 R_\xi^Z (g_2 \cos\theta_W + g_1 \sin\theta_W)^2,\quad 4 g_{Y'}^2 v_S^2 R_\xi^{Z'} \right\}\,.$$

(37)

# Acknowledgements

We would like to thank Kalliopi Petraki for useful comments on an earlier version of this manuscript. We would also like to thank Melissa van Beekveld for collaboration at an early stage of the project and Kalliopi Petraki, Andries Salm, and Adam Falkowski for useful discussions. We are supported by the NWO Vidi grant "Self-interacting asymmetric dark matter".

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
