# Peer review of "Whac-a-constraint with anomaly-free dark matter models"

_SciPost Physics, doi:SciPost Phys. 6, 020 (2019)_

## Round 2 · Referee Report · Anonymous (Referee 1) · 2018-11-7

Strengths

1- Detailed discussion of the importance of coannihilation and the sensitivty of electroweakino searches at the LHC 2- Clear presentation of the model under consideration and the mixing between the different fermions 3- Very instructive analysis of the impact of different LHC searches for different benchmark points

Weaknesses

1- Incomplete discussion of the upper bound on the masses of the various new particles 2- Missing constraints from direct detection

Report

The paper "Whac-a-constraint with anomaly-free dark matter models" considers minimal extensions of the Standard Model gauge group, constructed with the aim to provide a mediator for the interactions of dark matter. Phenomenological requirements on such models impose certain charge assignments (no couplings to leptons, mostly axial couplings to dark matter) that necessarily introduce gauge anomalies unless additional new states with Standard Model charges are present. The aim of the paper is to consider the phenomenological implications of these extra states, both for relic density calculations and for the LHC.

The present work continues an important ongoing discussion about the use and consistency of simplified models. By investigating the issue of anomaly cancellation, which has often been ignored in the literature, the paper provides a valuable addition to the literature. Although the specific model under consideration is not new, the paper provides important new phenomenology details. The presentation of the paper is clear and the results are of interest to the community. Once the comments below have been addressed, it will be suitable for publication.

Requested changes

Major changes:

1) While the paper makes the very important point that the new states needed for anomaly cancellation may have important phenomenological implications, it does not become clear to what degree these implications are unavoidable. Indeed, it has been argued that one can always assume the additional states to be heavy enough that the original simplified model is recovered at least approximately. It would be very useful to discuss this claim by reviewing the bounds on the masses of these new particles (e.g. by requiring perturbative Yukawa couplings). In fact, benchmark point [4] seems to have been chosen with exactly this goal in mind, so I would encourage the authors to discuss to what degree the phenomenology of this point can be captured in a simplified model approach.

2) I am somewhat puzzled by the choice of m_S for point [4]. For m_Z' = 240 GeV and g_Y' = 0.8, the vev of the extra Higgs boson is quite low, so the adopted value of m_S requires a very large quartic coupling, which appears to violate perturbative unitarity. Does the phenomenology change when reducing m_S to 1 TeV? If so, this would be a very interesting point (see comment 1).

3) The authors discuss direct detection constraints on both the spin-independent and the spin-dependent cross section. However, for spin-1 mediator models with dominantly axial couplings to DM and dominantly vector couplings to quarks, the strongest constraints may in fact come from momentum-suppressed (but still spin-independent) interactions. PandaX has recently published bounds on this operator (called L^13 in arXiv:1807.01936), so the authors should check whether all benchmark points are consistent with these constraints.

4) The authors limit their analysis to the case that DM is a Dirac fermion. It would be good to add at least a qualitative discussion of how the discussion would change for a Majorana fermion. Would it still be possible to achieve anomaly freedom with the same field contents? If not, what modifications are necessary? Would such a model suffer from less tuning (given that the vector coupling to DM is forced to vanish)?

Minor changes:

5) Although very clear, I find the notation for the mixed Yukawa couplings extremely hard to read. Given that all of these Yukawa couplings involve \phi and that they all involve one left- and one right-handed field, the index \phi_L and \phi_R is in fact redundant, i.e. one could simply name them y_{\theta_R} etc. To avoid a clash with the Majorana mass terms for \chi, the latter could receive an additional label (e.g. a superscript M). Since the Majorana mass terms are anyways set to zero for most of the paper, the remaining notation should then be much easier to read.

6) If the authors insist on keeping the current title, there should be a brief explanation of it in the introduction. Also, I would encourage the authors to provide an outline of the paper.

  • validity: high
  • significance: high
  • originality: good
  • clarity: high
  • formatting: good
  • grammar: excellent

Author:  Karl Nordström  on 2019-01-24  [id 410]

(in reply to Report 1 on 2018-11-07)
Category:
correction

We would like to thank the referee for reading our manuscript carefully and providing many useful remarks which have allowed us to improve the overall quality of the work. We respond to the requested changes below:

1) We have included a discussion of the upper bounds on the masses of the additional states in an updated version of the manuscript. Combining relic density bounds with perturbative unitarity requirements in the Yukawa sector leads to a limit of the order of tens of TeV for the heavy fermion masses. This bound sits firmly outside of the LHC energy range but could possibly be probed by a 100 TeV collider. This demonstrates that the original simplified model is a valid limit of the theory at least at the LHC.

2) This value of m$_S$ indeed requires a non-perturbative value for the quartic coupling, we thank the referee for pointing out this inconsistency. As the new scalar is effectively decoupled from the collider phenomenology (unless is it light enough to be accessible through decays), changing its mass to $1$ TeV can only have an effect only on the relic density, and we have confirmed that this effect is negligible.

3) We thank the referee for pointing out the new PandaX-II bounds which are considerably stronger than those previously set on the momentum-suppressed operator. Indeed parameter points 1,2,4 violate these constraints. We have made a new plot demonstrating our recasting of the PandaX-II constraints and added a discussion to reflect their impact. We have also generated some new parameter points that are allowed under these constraints. We keep the violating points in the manuscript but present them as non-thermal parameter points which can not explain the full dark matter relic density, but demonstrate possible collider signatures.

4) Turning on both Majorana masses indeed splits the two Dirac mass eigenstates in the neutral fermion sector into four Majorana fermions and hence forces the vectorlike couplings to be off-diagonal, which avoids the extreme fine-tuning of the spin-independent direct detection cross section through the $Z$ (assuming the splitting is large enough to kinematically disallow it). There are still constraints on the mixing angle, as both the spin-dependent and momentum-suppressed spin-independent contributions through the $Z$ remain, but these are of course much less stringent. The dark matter and collider phenomenology is even richer than in the Dirac-case as the model now involves four Majorana fermions and one charged Dirac fermion in the limit where the doublet is not much heavier than the singlets. A detailed study of this scenario would be interesting, however its phenomenology should share many qualitative features with the Dirac limit we have studied, with possible longer cascade decays. We have added a discussion along these lines to the manuscript.

5) We have updated the manuscript to use this (improved) notation.

6) We have added a brief explanation of the title and an outline of the paper to the introduction.

---

## Round 2 · Referee Report · Anonymous (Referee 2) · 2018-12-7

Strengths

1- Written in a clear and (largely) self-consistent way 2- Relevant contribution to the bottom-up approach to dark matter phenomenology

Weaknesses

1- The study does not go far beyond existing literature, in particular Ref. [19]

Report

This paper presents a phenomenological study of an anomaly-free dark matter model with a vector mediator. Evading strong constraints from the LHC and direct detection, respectively, the authors consider a leptophobic mediator that couples axially to dark matter. The authors point out the predictivity of the requirement of anomaly cancellation providing a rich phenomenology. They compute the relic density, particularly considering regions of co-annihilations, and impose constraints from direct detection and the LHC. A study of indirect detection constraints and an exploration of the parameter space beyond a number of benchmark points is left for future work. One of the main conclusions of the paper, going beyond the existing literature on the subject, is the importance of SUSY LHC searches to constrain the model. The topic of the paper is relevant and likely to be of wide interest. It is clearly written and well structured. I have, however, two requests to be address before recommending the paper for publication in SciPost (see below).

Requested changes

1- The subject of the paper is very similar to the one of Ref. [19]. The authors should make this clear and point out their novel contributions in the introduction. I propose to add this information in the last paragraph of the introduction.

2- As the authors correctly point out in Sec. 3.1, the efficiency of the interconversion rates during dark matter freeze-out in the presence of co-annihilation is questionable in the considered model. The authors (implicitly) require these rate to be an order magnitude larger than the Hubble rate. Looking e.g. at Fig. 1 of 1705.09292 this does not seem to always be sufficient. The authors should, hence, discuss this more clearly. BTW, I assume the authors mean that the "(interconversion) rate is only an order of magnitude larger than the Hubble rate" (the cross section itself cannot be compared to the Hubble rate).

  • validity: high
  • significance: high
  • originality: good
  • clarity: top
  • formatting: excellent
  • grammar: excellent

Author:  Karl Nordström  on 2019-01-24  [id 411]

(in reply to Report 2 on 2018-12-07)
Category:
correction

We would like to thank the referee for carefully reading our manuscript and providing helpful comments, which allowed us to find a mistake in the interconversion rate calculation. We respond to the requested changes below:

1) We have added a sentence discussing the differences to Ref. [19] to the introduction in an updated version of the manuscript.

2) There was a mistake in our original calculation of the interconversion rate as we had left out diagrams with a $t$-channel $Z$. With these diagrams included, the interconversion rate is several orders of magnitude above the Hubble rate for dark matter masses around the weak scale. The large difference is caused by being able to scatter off effectively massless quarks in the plasma rather than only particles with weak-scale masses. We have included an expanded discussion of how the interconversion rate is calculated in the updated manuscript.

---

## Round 2 · Referee Report · Tilman Plehn (Referee 3) · 2019-1-6

Strengths

The paper is on top of the LHC discussions and technically as well as phenomenologically very sound. Its point that simplified models are poor guides to LHC searches should definitely be made and published.

Weaknesses

Of course, there is no chance that the LHC will discover new physics
based on a model suggested by phenomenologists, especially if the
model's rather baroque structure is determined purely by the need to avoid experimental constraints. But I like the way that the paper shows how simplified models for mono-jet searches are completely misleading, but this story should be told a little more clearly.

Report

The authors introduce a dark model with an axial-vector mediators not coupling to leptons, to avoid many of the existing constraints. They discuss the additional particles in some detail, analyze their role in thermal dark matter production, and provide the current LHC constraints. Their main result is that for most of their benchmark points mono-jet searches do not provide the strongest constraints, even if the corresponding simplified model is constructed as leptophobic.

Requested changes

1- At the very beginning the authors claim that simplified models are not self-consistent. It would be good to argue more specifically that they are constructed at tree level and are not a well-defined quantum field theory. From that perspective I also disagree with the bottom-up and top-down argument, because even bottom0up Lagrangians need to be quantized.

2- In the introduction on p.2 the authors motivate a certain class of models. Now, at least almost all models suggested by theorists will eventually be proven wrong, so it would be good to argue that this paper is not about an interesting model but about showing the absurdity of simplified models.

3- I am missing a little bit more of a structure in the discussion of the model starting on p.3. It might be helpful to give a graphic representation of a couple of sample spectra.

4- On p.5, why is an additional U(1) symmetry a reason to ignore Majorana mass terms?

5- On p.6, could the authors please comment on scalar mixing beyond tree level? What happens to the analysis if we allow for a small mixing?

6- Why is it reasonable to assume zero kinetic mixing at some high scale? I would agree with such an argument if the model had a GUT-like motivation, but this argument is never given before. For the running of the mixing, what are the assumptions of the new particle masses/thresholds?

7- Before going into Sec. 3.1 it would be nice to mention that the authors consider the relic density a constraining factor. I agree with this approach, but some of our colleagues consider this an unjustified relic density fetish.

8- On p.8, I am not sure every reader is familiar with the procedure applied in Ref.[64,65], please be more specific.

9- I am not sure I understand the argument concerning the chi-chi mixing, especially how strongly we can argue that the mixing remains small even for equal masses.

10- Still on p.10, given the relevance of W and Z funnels in the co-annihilation, are the Feynman diagrams in Fig.3 really the ones the authors would like to show?

11- Starting on p.12 please add a few Feynman diagrams with the relevant LHC processes, including the new particles, etc. As it stands it is very hard to get a grip on the different LHC constraints (which I have no doubt are correctly computed).

12- For a quantitative discussion of electroweakino pair production in supersymmetric models it might be helpful to look into our recent paper on exactly this topic (I am not aware of other focused studies of this topic). At least older readers would appreciate some more analogies with the established supersymmetric models.

13- Sorry, but I am not sure I understand the argument on p.13 as to why the analysis is conservative, even in the presence of (neglected) systematic and theory uncertainties.

14- On p.14, where do these benchmark points come from and what kind of patterns are they expected to represent? Where do they stand in terms of non-LHC constraints? And why are the three first benchmark points most sensitive to the same analysis looking for soft OS-leptons? The fact that these leptons are soft suggests a link to small mass splittings?

15- At the end, the paper seems to end without making an explicit point of the funny bottom line. And what would remain of this bottom line when we loosen the relic density constraint? Would a too small relic density have an effect on the most relevant LHC searches?

16- There are a few minor language problems/typos, please read the paper once more carefully.

17- I know it is a pain, but the reference list misses a lot of relevant references on anomalies related to Z' searches.

18- Please update the references with the journal numbers. Actually, I only care about the SciPost references, for example our Ref.[17].

  • validity: high
  • significance: high
  • originality: high
  • clarity: ok
  • formatting: good
  • grammar: excellent

Author:  Karl Nordström  on 2019-01-24  [id 412]

(in reply to Report 3 by Tilman Plehn on 2019-01-06)
Category:
answer to question
correction

We would like to thank the referee for their detailed and helpful comments. We respond to their points in detail below:

1) We have rephrased the first paragraph of the introduction in an updated version of our manuscript to make it clear that simplified models that do not generate the masses of the $Z'$ and the dark matter in a consistent manner and/or do not cancel gauge anomalies cannot be quantised and therefore cannot be considered realistic models of nature in any meaningful way.

2) As the original simplified model is still a consistent limit of our model within the LHC energy range (we have included an expanded discussion around how high the masses of the new fermions can be pushed), we believe that such a strong statement would not be warranted. However we have added a sentence to the introduction which reflects how our findings highlight the need to take constraints from simplified model studies with a grain of salt.

3) We have added a figure which displays the mass spectra of two benchmark points: one with significant mass mixing in the fermion sector (which therefore is ruled out by direct detection), and one with minimal mixing which corresponds to one of the benchmark points for the LHC analysis.

4) We agree that motivating the absence of the Majorana terms with a spurious symmetry is not warranted. We have changed the discussion around the Majorana terms to reflect that the Dirac limit can be motivated by simplicity. Working in this limit reduces the number of physical parameters that one needs to take into account phenomenologically while still capturing the qualitative features of the relic density and collider phenomenology of the full model (moreover, turning on small Majorana terms will suppress the constraints on the mass mixing with the doublet component in the neutral fermion sector).

5) Higher order corrections will of course induce a small mixing in the scalar sector even when the mixing vanishes at tree level. While spin-independent interactions through the $125$ GeV scalar are quite constrained by direct detection, they are not nearly as constrained as interactions taking place through the $Z$: for the operator $y H \bar{\chi} \chi$, $y \lesssim 0.1$. We have added a short discussion along these lines to the manuscript.

6) We have clarified that gauge unification is only one way to fix the kinetic mixing parameter and that we in principle are free to set it to $0$ at any scale we wish. However it is a well-motivated reason to study the effect of setting it to 0 at a higher scale in the model, as we do. Since only the quarks contribute to the running of the kinetic mixing parameter we do not have to worry about thresholds unless we go below the top mass.

7) We have added a sentence to make this clear at the beginning of the section.

8) We have added some details about this calculation to the manuscript.

9) We have changed the wording slightly. The intention was to say that the direct detection constraint on the mixing angle (which is independent of the masses themselves but rather just leads to fine-tuning the mixing terms in the lagrangian) still allows for prompt decays of the heavy fermions as long as the mass difference between them and their products is larger than about $10$ GeV. For smaller mass differences, long-lived particle searches would have to be considered.

10) The diagrams involving the new scalar are important for understanding many of the features of the region where $m_{\chi_1} > m_{Z'}/2$ in Figure 5, and the diagrams showing the interconversion processes are important to understand this calculation. We have added an extra row of example diagrams which shows co-annihilation processes into weak bosons to illuminate this feature of the model.

11) We have added some representative diagrams of pair production of the heavier fermions and their decays.

12) The collider phenomenology of our model is similar to the Dirac bino - higgsino limit of the MSSM, with the addition of the $Z'$ which allows for more efficient $s$-channel pair production of neutralinos and charginos (in the U1MSSM is that the typical charge assignment is not leptophobic, which forces $m_{Z'} \gtrsim 3$ TeV). We have added a comment along these lines and a citation to Bernreuther:2018nat, which indeed discusses similar collider searchers in the MSSM and NMSSM.

13) We agree that this wording is not warranted and have removed it.

14) The benchmark points are selected to provide the correct relic density while avoiding all current constraints, and have sensitivity to a wide range of LHC searches, as specified in the text. The points are detailed in Table 1 and the first three are indeed coannihilation scenarios with small mass splittings (as these are well-motivated from the perspective of the relic density). As detailed in Table 1 they decay to relatively soft three-body final states, which makes it unsurprising that the most sensitive searches are soft OS lepton pair + MET searches, as e.g. $\chi_2\, \chi_2 \rightarrow \ell^+ \ell^- q\,\bar{q} \chi_1 \chi_1$ will in general lead to one or two jets (depending on the $Z$ boost), MET, and a low mass OS lepton pair. However there are qualitative differences between the CMS and ATLAS selection cuts, which is why point 1 is more sensitive to the CMS analyses and point 3 to the ATLAS analysis: the ATLAS analysis requires a hard jet ($p_T > 100$ GeV) in addition to the low-mass OS lepton pair, which preferentially selects scenarios where $m_{Z'} \gg 2 m_{\chi_2}$ such that the $Z$ decays products often are boosted and the hadronic decay can be reconstructed as a single jet, while CMS also puts an upper limit on the allowed $p_T < 50$ GeV of the leptons, and thus tends to reject such events.

15) This is perhaps a cultural difference in senses of humor, as the authors prefer the more understated nature of the current text to spelling out the joke. We hade added a sentence which discusses the effect of loosening the relic density constraint to the results section -- in short, this means we require a larger g_{Y'} for a parameter point, everything else being equal (assuming Yukawa couplings and the scalar quartic remain perturbative), which will in general lead to even more sensitivity to LHC searches.

16) We have corrected all the language mistakes we have found.

17) We have added citations to a number of relevant papers studying the phenomenology of anomaly-free $U(1)$ models of $Z'$ in various contexts to the introduction.

18) We have updated the references.

Tilman Plehn  on 2019-01-27  [id 415]

(in reply to Karl Nordström on 2019-01-24 [id 412])
Category:
remark

Thanks for all the clarifications! I still think you are hiding a really funny story relatively deep in that paper, and your dry and of course exquisite sense of humor will prevent lots of potential readers from getting it (and citing you for it). But then who am I to teach others sense of humor...

---

## Editorial Decision

published